# Group III metabotropic glutamate receptors gate long-term potentiation and synaptic tagging/capture in rat hippocampal area CA2

Ananya Dasgupta[1,2], Yu Jia Lim[1], Krishna Kumar[1,2†], Nimmi Baby[1,2], Ka Lam Karen Pang[1,2], Amrita Benoy[1,2], Thomas Behnisch[3], Sreedharan Sajikumar[1,2]*

[1]Department of Physiology, Yong Loo Lin School of Medicine, National University of Singapore, Singapore, Singapore; [2]Life Sciences Institute Neurobiology Programme, National University of Singapore, Singapore, Singapore; [3]Institutes of Brain Science, State Key Laboratory of Medical Neurobiology and MOE Frontiers Center for Brain Science, Fudan University, Shanghai, China

**Abstract** Metabotropic glutamate receptors (mGluRs) play an important role in synaptic plasticity and memory and are largely classified based on amino acid sequence homology and pharmacological properties. Among group III metabotropic glutamate receptors, mGluR7 and mGluR4 show high relative expression in the rat hippocampal area CA2. Group III metabotropic glutamate receptors are known to down-regulate cAMP-dependent signaling pathways via the activation of $G_{i/o}$ proteins. Here, we provide evidence that inhibition of group III mGluRs by specific antagonists permits an NMDA receptor- and protein synthesis-dependent long-lasting synaptic potentiation in the apparently long-term potentiation (LTP)-resistant Schaffer collateral (SC)-CA2 synapses. Moreover, long-lasting potentiation of these synapses transforms a transient synaptic potentiation of the entorhinal cortical (EC)-CA2 synapses into a stable long-lasting LTP, in accordance with the synaptic tagging/capture hypothesis (STC). Furthermore, this study also sheds light on the role of ERK/MAPK protein signaling and the downregulation of STEP protein in the group III mGluR inhibition-mediated plasticity in the hippocampal CA2 region, identifying them as critical molecular players. Thus, the regulation of group III mGluRs provides a conducive environment for the SC-CA2 synapses to respond to events that could lead to activity-dependent synaptic plasticity.

*For correspondence:
phssks@nus.edu.sg

Present address: [†]School of Advanced studies, University of Tyumen, Tyumen, Russia

Competing interests: The authors declare that no competing interests exist.

## Introduction

Synaptic plasticity and memory formation have remained fascinating areas of neuroscience research and have been widely studied in the hippocampus, a brain area that is primarily responsible for memory formation and spatial navigation. The hippocampal area CA2 has recently been the subject of much attention owing to its involvement in social memory formation and socio-cognitive information processing (*Hitti and Siegelbaum, 2014*; *Pagani et al., 2015*; *Smith et al., 2016*). The excitatory neurons in the CA2 area have unique intrinsic properties such as more negative resting membrane potentials (*Zhao et al., 2007*). Distal CA2 neurons receive strong excitatory inputs from layer II of the entorhinal cortex (EC-LII) directly and these EC-CA2 synapses express activity-dependent long-term potentiation (LTP) (*Chevaleyre and Siegelbaum, 2010*). In contrast, the Schaffer collateral inputs (SC) from CA3 to CA2 do not support activity-dependent LTP in response to typical induction protocols (*Zhao et al., 2007*). Although the functional importance for this difference in

synaptic plasticity is unclear, it reflects the ability of various brain regions to precisely regulate activity-dependent modulation of synaptic efficiency (*Caruana et al., 2012*; *Dudek et al., 2016*). In contrast to adjacent hippocampal areas, CA2 has a distinct molecular profile, with an enhanced expression of several genes and proteins, such as Striatal-enriched protein tyrosine phosphatase (STEP) (*Boulanger et al., 1995*; *Zhao et al., 2007*; *Lee et al., 2010*; *Caruana et al., 2012*; *Dudek et al., 2016*; *Farris et al., 2019*) and metabotropic glutamate receptor 4 (mGluR4), a specific group III mGluR (*Fotuhi et al., 1994*; *Ohishi et al., 1995*; *Phillips et al., 1997*).

Group III metabotropic glutamate receptors are G-protein coupled receptors which consists of four subtypes, namely mGluR4, mGluR6, mGluR7 and mGluR8. They are largely presynaptically localized and downregulate neurotransmitter release from presynaptic terminals directly or indirectly (*Niswender et al., 2010*). These receptors are negatively coupled to adenylyl cyclase and, via the activation of $G_{i/o}$ proteins, lead to the inhibition of the cAMP cascade that is critical for the maintenance of long-term synaptic plasticity (*Mercier and Lodge, 2014*). Therefore, given the prominent expression of mGluR4 in the CA2 region of the hippocampus (*Phillips et al., 1997*), we speculated that the elevated expression of group III mGluRs in the CA2 area is one of the factors contributing to the resistance to canonical activity-dependent plasticity observed at the SC-CA2 synapses.

A typical mGluR is primarily composed of an intracellular C-terminal tail, seven transmembrane domains (heptahelical) and an extracellular N-terminal domain or Venus flytrap domain (VFD), which 'traps' orthosteric ligands (*Beqollari and Kammermeier, 2010*; *Niswender et al., 2010*). In addition, a cysteine-rich linker bridges the N-terminal domain with the first of seven transmembrane domains. Although a number of orthosteric agonists that are specific to each mGluR has been identified to date, mGluR specific antagonists are relatively scarce. Orthosteric antagonists such as (RS)-MPPG and (RS)-CPPG are selective to group III mGluRs and display a hierarchical preference to these receptors over group II mGluRs. (RS)-a-cyclopropyl-4-phosphonophenyl glycine or CPPG, a phenylglycine analogue, shows a 30-fold selectivity towards group III receptors over group II receptors (*Miller et al., 2003*). In order to develop a more potent group III mGluR antagonist, derivatives of the parent phenylglycines (MPPG and CPPG) were produced, such as UBP1110, UBP1111 and UBP1112 (*Miller et al., 2003*). These compounds also exhibit greater selectivity towards group III mGluRs over group II mGluRs.

In this study, we aimed to investigate in greater detail the expression and role of group III mGluRs in the hippocampal CA2 region. We analyzed the relative expression levels of different group III mGluRs within the CA2 region and hypothesized that the high expression of group III mGluRs in area CA2 is a probable factor limiting plasticity at the Schaffer collateral-CA2 synapses. Due to the lack of specific antagonists for each group III mGluR subtype, we used two general but potent group III mGluR antagonists, (RS)-CPPG and UBP1112, and studied if activity-dependent plasticity could be induced in the LTP-resistant Schaffer collateral-CA2 synapses upon group III mGluR inhibition. We further addressed whether group III mGluR inhibition supports late associativity, a unique phenomenon that leads to the strengthening of weakly stimulated synaptic inputs that are usually incapable of expressing long-lasting potentiation. This study also highlights the activation of ERK/MAPK and the downregulation of STEP protein in area CA2 as a result of group III mGluR inhibition.

## Results

### Pharmacological inhibition of group III mGluRs promotes NMDA receptor- and protein synthesis-dependent long-lasting LTP in Schaffer collateral-CA2 synapses

Schaffer collateral-CA2 synapses are known from previous studies to be resistant to activity-dependent LTP induced by conventional induction protocols that are effective in CA1 neurons (*Zhao et al., 2007*). Here, we showed that, with basal stimulation, CA2 synapses displayed stable responses (*Figure 1B*, Wilcoxon test; p>0.05 at any given time point) and confirmed that Schaffer collateral-CA2 synapses did not express LTP after triple 100 Hz tetanizations (STET; *Figure 1C*; blue circles; Wilcoxon test; p>0.05 at any given time point). In contrast, entorhinal cortical-CA2 synapses expressed activity-dependent late-LTP when stimulated with the same tetanization protocol (*Figure 1D*; red circles; Wilcoxon test; p=0.0113 at 60, 120, 180 and 240 min).

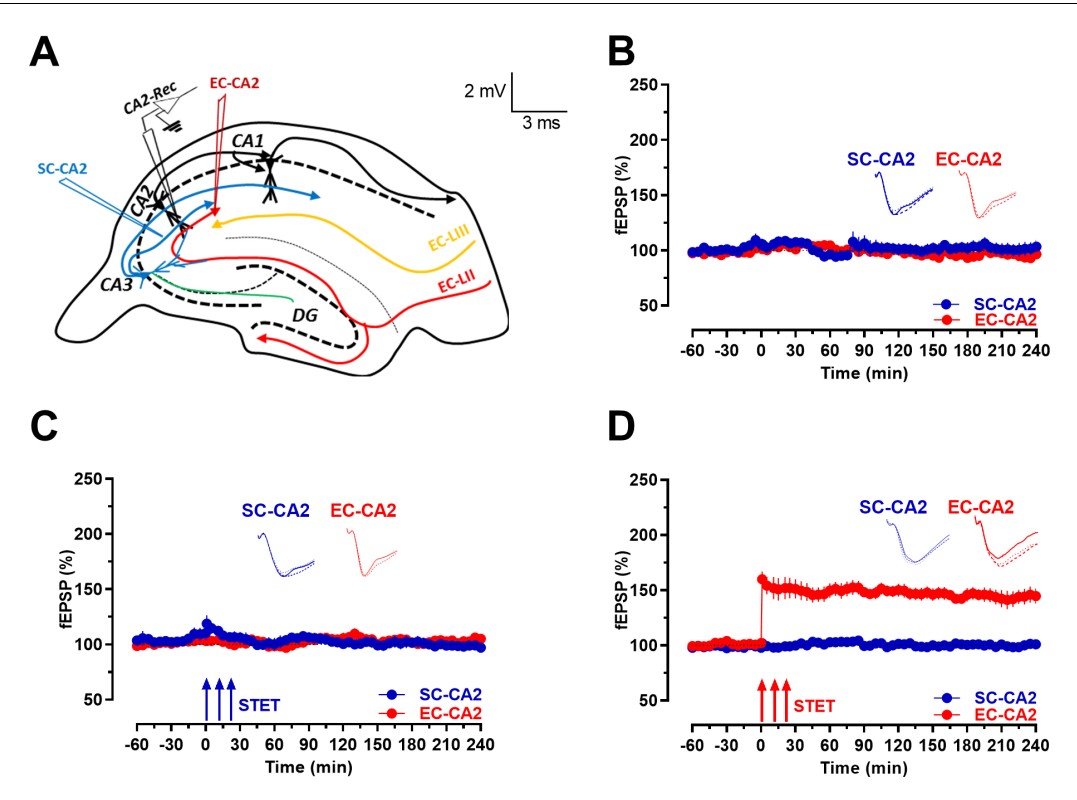

**Figure 1.** Schaffer collaterals to CA2 neurons are resistant to activity-dependent long-term potentiation while the entorhinal cortical synapses onto CA2 express late-LTP. (A) Schema showing location of stimulating and recording electrodes. SC-CA2 (blue) and EC-CA2 (red) stimulating electrodes and recording site (black) of fEPSPs within the hippocampal CA2 area are shown. (B) fEPSP recordings under baseline stimulation recorded from SC-CA2 (blue circles) and EC-CA2 (red circles) synaptic inputs for more than 4 hr indicate their stability over time (n = 7). (C) STET at SC-CA2 did not lead to expression of LTP (blue circles; n = 7). EC-CA2 inputs under baseline stimulation exhibited stable potentials throughout the experiment (red circles). (D) STET at EC-CA2 leads to expression of late-LTP (red circles; n = 6) lasting up to 240 min. SC-CA2 inputs under baseline stimulation exhibited stable fEPSPs (blue circles). Representative fEPSP traces 30 min before (closed line), 60 min after (dotted line), and 240 min after (hatched line) STET are depicted. Calibration bars for fEPSP traces in all panels are 2 mV/3 ms. Arrows indicate the time points of STET. 'n' represents number of slices.

We hypothesized that this plasticity resistance property of CA2 synapses could be attributed to its unique expression of surface receptors. In particular, the spatial expression pattern of mGluR4 is distinctly concentrated in area CA2, in contrast to most of the other mGluRs that are widely expressed throughout the rat hippocampal formation (*Fotuhi et al., 1994*). mGluR4 expression is the highest in the area CA2, while there is only sparse mGluR4 expression in the dentate gyrus, CA3, and subiculum, and no observable expression in the CA1 region. A similar trend has also been reported in other studies (*Ohishi et al., 1995*; *Phillips et al., 1997*).

To corroborate the above observations, we further studied the relative expression levels of group III mGluRs within the CA2 region using qPCR. We observed that mGluR7 and mGluR4 mRNA expression levels within the CA2 region were much higher than the expression of mGluR6 and mGluR8 (*Figure 2A*, one-way ANOVA, p=0.0001). The high relative expression levels of mGluR7 and mGluR4 within area CA2 point towards a possible role of these receptors in restricting the induction of synaptic plasticity in the SC-CA2 synapses. To study if pharmacological inhibition of group III mGluRs promotes LTP induction at SC-CA2 synapses, we bath applied group III mGluR antagonists, either (RS)-CPPG or UBP1112, for a total time period of 1 hr. After the initial 30 min of drug application, a strong tetanization (STET, triple 100 Hz tetanizations) was given to the SC-CA2 synaptic input to induce late-LTP in the presence of the antagonists. When either (RS)-CPPG (*Figure 2B*; blue circles) or UBP1112 (*Figure 2C*; blue circles) was bath applied, STET in SC-CA2 input induced an immediate significant LTP (at 30 min, Wilcoxon test; p=0.005 and 0.017 in *Figure 2B* and *Figure 2C*, respectively) that was maintained for at least 4 hr (at 240 min, Wilcoxon test; p=0.013 and 0.027 in

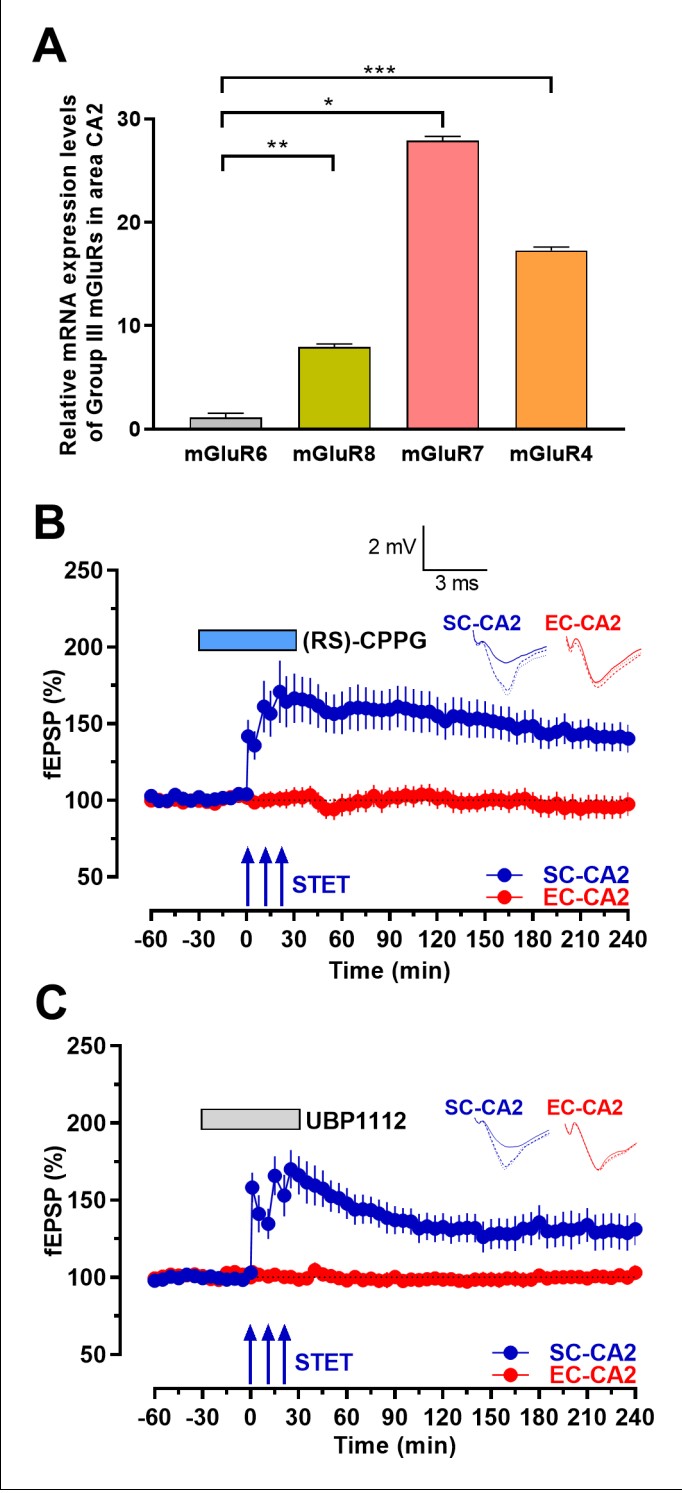

**Figure 2.** Inhibition of group III mGluRs leads to expression of activity-dependent late-LTP at the Schaffer collaterals to CA2 synapses. (**A**) mRNA expression levels of group III mGluRs in area CA2. qRT-PCR analysis showing significantly higher expression levels of mGluR7, mGluR4 and mGluR8 compared to mGluR6. Significant difference between the groups mGluR6 vs mGluR8, mGluR7 and mGluR4 are indicated by * $P < 0.05$, ** $P < 0.01$ and *** $P < 0.001$ (one-way ANOVA, 12 slices each from four different biological samples, n = 4; 'n' represents number of animals.) (**B**) Group III mGluR antagonist (RS)-CPPG (1 μM) was bath applied for 1 hr after recording a stable baseline of 30 min. 30 min into (RS)-CPPG application, STET was delivered at SC-CA2 inputs, which resulted in late-LTP lasting 4 hr at SC-CA2 (blue circles; n = 11). (**C**) Group III mGluR antagonist UBP1112 (15 μM) was bath

*Figure 2 continued on next page*

*Figure 2 continued*

applied for 1 hr after recording a stable baseline of 30 min. 30 min into UBP1112 application, STET was delivered at SC-CA2 inputs, which resulted in late-LTP lasting 4 hr at SC-CA2 (blue circles; n = 7). EC-CA2 inputs (red circles) exhibited stable fEPSPs throughout the recording period after antagonist application (C, D). Horizontal bars indicate drug application period. Representative fEPSP traces 30 min before (closed line), 60 min after (dotted line), and 240 min after (hatched line) STET are depicted. Calibration bars for fEPSP traces in all panels are 2 mV/3 ms. Arrows indicate the time points of STET. 'n' represents number of slices in the electrophysiology experiments.

*Figure 2B* and *Figure 2C*, respectively). Stable baseline responses were recorded throughout the experimental period in the cortical input in both cases (*Figure 2B & C*; red circles; p>0.05 at any given time point). The observed potentiation was also significantly different between both SC- and EC-CA2 synaptic inputs (at 240 min, Mann-Whitney test; p=0.02 and 0.038 in *Figure 2B* and *Figure 2C*, respectively). Stable, unaltered baseline responses were recorded throughout the experimental period in the cortical input in both cases (*Figure 2B & C*; red circles; p>0.05 at any given time point), indicating that these drugs did not have the general effect of increasing synaptic transmission.

Additionally, we performed whole-cell voltage-clamp recordings of single CA2 pyramidal neurons under drug-free conditions and in the presence of either (RS)-CPPG or UBP1112. We observed no changes in EPSCs in response to control stimulation, with or without antagonists, for the entire length of the recordings (*Figure 3A,C,D*), validating that these pharmacological compounds did not have non-specific effects on the baseline control EPSCs. Also, paired HFS evoked only a decaying potentiation of synaptic transmission, lasting less than 10 min (*Figure 3B*, Wilcoxon test; p=0.625 at 10 min), confirming the lack of long-lasting LTP in SC-CA2. To study the effect of the drugs on synaptic potentiation in single CA2 cells, we measured SC-CA2 evoked EPSCs before and after paired HFS. Application of the mGluR antagonists resulted in a statistically significant synaptic potentiation immediately after paired HFS (*Figure 3E & F*, Wilcoxon test; p=0.0156 and 0.0156), lasting for the entire length of the recording.

The synaptic localization of group III mGluRs is predominantly presynaptic (*Niswender and Conn, 2010*), whereas the induction and expression of classical LTP is postsynaptic (*Nicoll, 2003*; *Kerchner and Nicoll, 2008*). Thus, we tested whether group III mGluR inhibition facilitates LTP in SC-CA2 by acting presynaptically. To test this, we have analyzed the effect of group III mGluR antagonists (RS)-CPPG or UBP1112 on high frequency stimulation-mediated summation of field potentials (*Figure 3—figure supplement 1A–F*) and high frequency stimulation-mediated EPSCs at 0 mV holding potential recorded in the whole-cell voltage clamp configuration (*Figure 3—figure supplement 2A–C*). In both cases, we could not observe any presynaptic changes. Thus, in our experimental conditions, the observed SC-CA2 LTP is not based on mechanistically enhanced glutamate release during the 100 Hz trains. We therefore assume, together with the other data presented in this study, that the observed effect of the group III mGluR inhibition is postsynaptically mediated. In addition, when a weak tetanization was delivered to EC-CA2 and SC-CA1 synaptic inputs in the presence of (RS)-CPPG or UBP1112, a short-lasting early-LTP was observed in the tetanized pathways, similar to that in control conditions (*Figure 3—figure supplement 3A–H*). These results suggest that the LTP induction threshold was not changed by group III mGluR inhibition in the EC-CA2 and SC-CA1 pathways. Hence, application of group III mGluR inhibitors lowers LTP induction threshold only in SC-CA2 synapses.

To further understand the nature of group III mGluR inhibition-mediated LTP, we analyzed its protein synthesis and NMDA receptor dependency using protein synthesis inhibitors emetine and anisomycin, and the NMDA receptor antagonist D-AP5. After obtaining a stable baseline for at least half an hour, anisomycin (25 μM; *Figure 4A & C*) or emetine (20 μM; *Figure 4B & D*) were co-applied with the group III mGluR antagonists (RS)-CPPG or UBP1112. Synaptic potentiation of SC-CA2 synaptic input was significantly lower immediately after STET in response to the co-application of anisomycin (*Figure 4A and C*, Wilcoxon test; p=0.017 and 0.035) or emetine (*Figure 4B and D*, Wilcoxon test; p=0.020 and 0.018) with group III mGluR antagonists (RS)-CPPG (*Figure 4A and B*) or UBP1112 (*Figure 4C and D*). Statistically significant potentiation in *Figure 4A* after STET was maintained for only 60 min (Wilcoxon test; p=0.042), while the potentiation in *Figure 4C* was maintained for only 150 min (Wilcoxon test; p=0.035). In the case of emetine (*Figure 4B and D*), statistically significant

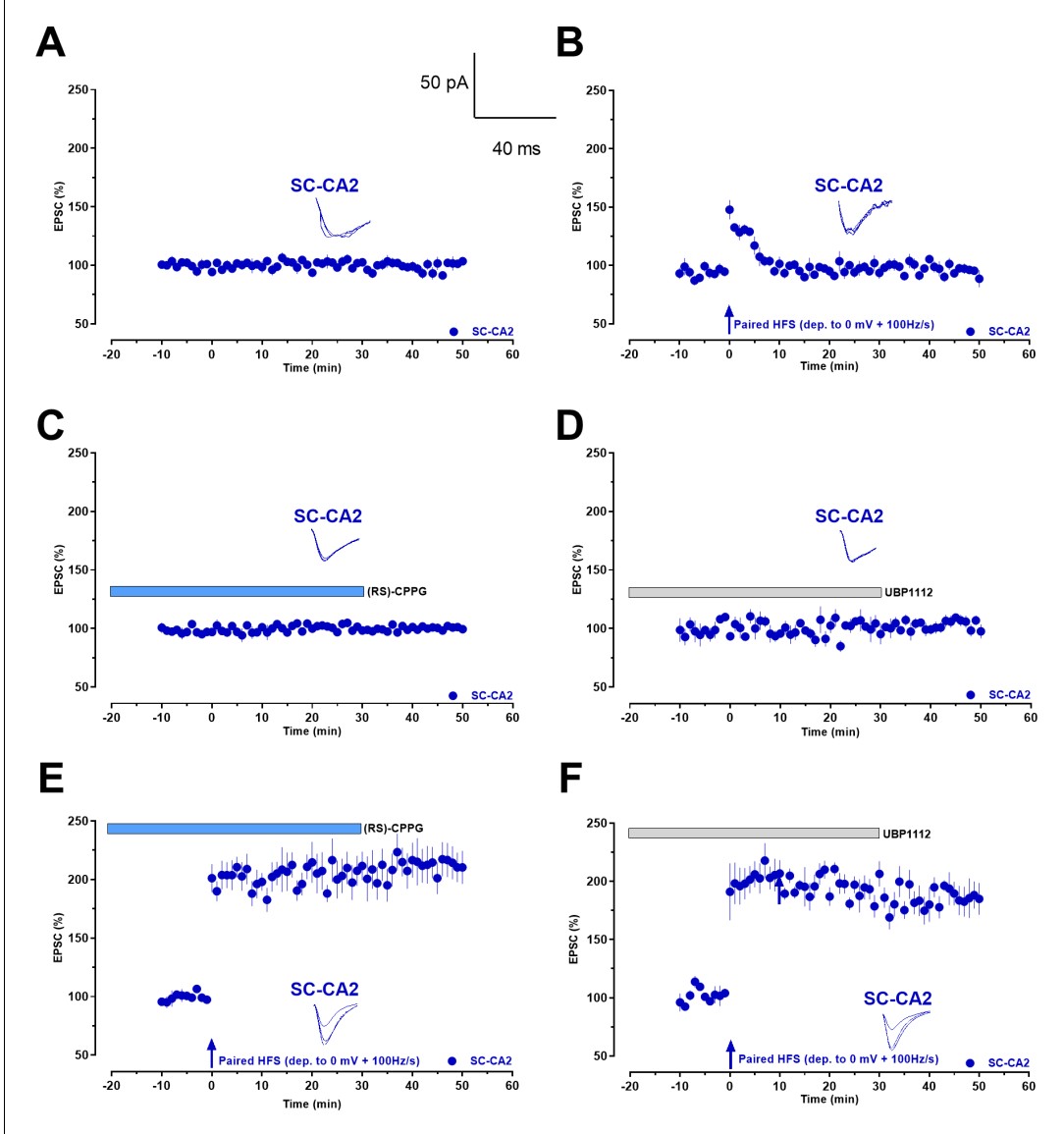

**Figure 3.** Whole-cell voltage-clamp recordings demonstrate that group III mGluR inhibition leads to activity-dependent late-LTP at Schaffer collaterals to CA2 synapses in single cells. (**A**) Control experiment with evoked EPSCs recorded from CA2 pyramidal neurons under basal stimulation of Schaffer collaterals shows the stability of the whole-cell recordings (n = 5). (**B**) High frequency stimulation (HFS) at Schaffer collaterals paired with a membrane depolarization to 0 mV (paired HFS (dep. to 0 mV + 100 Hz/s)) after a 10 min baseline recording did not cause an expression of LTP at CA2 pyramidal neurons (n = 7) in the absence of group III mGluR antagonists and EPSCs decayed back to baseline quickly. (**C**) Bath application of (RS)-CPPG for a total period of 50 min under baseline SC-CA2 stimulation resulted in stable EPSCs throughout the recording period (n = 7). (**D**) Bath application of UBP1112 for a total period of 50 min under baseline SC-CA2 stimulation resulted in stable EPSCs throughout the recording period (n = 5). (**E**) (RS)-CPPG was bath applied for a total time period of 50 min (20 min before and 30 min after HFS) and HFS resulted in significant potentiation of EPSCs at SC-CA2 (n = 5). (**F**) UBP1112 was bath applied for a total time period of 50 min (20 min before and 30 min after HFS) and HFS resulted in significant potentiation of EPSCs at SC-CA2 (n = 5). Vertical blue arrows: time point of paired HFS. Horizontal bars: drug application period. Representative EPSC traces 5 min before (closed line), 20 min after (dotted line), and 50 min after (hatched line) paired HFS are depicted. Calibration bars for EPSC traces in all panels are 50 pA/40 ms. 'n' represents number of slices.

The online version of this article includes the following figure supplement(s) for figure 3:

**Figure supplement 1.** Effect of (RS)-CPPG or UBP1112 on high frequency stimulation mediated summation of field potentials.

**Figure supplement 2.** Effect of (RS)-CPPG and UBP1112 on high frequency stimulation-mediated EPSCs at 0 mV holding potential recorded in the whole-cell voltage-clamp configuration.

**Figure supplement 3.** Bath application of group III mGluR inhibitors does not lower LTP induction thresholds in the Schaffer collateral (SC)-CA1 synaptic inputs and entorhinal cortex (EC)-CA2 synaptic inputs.

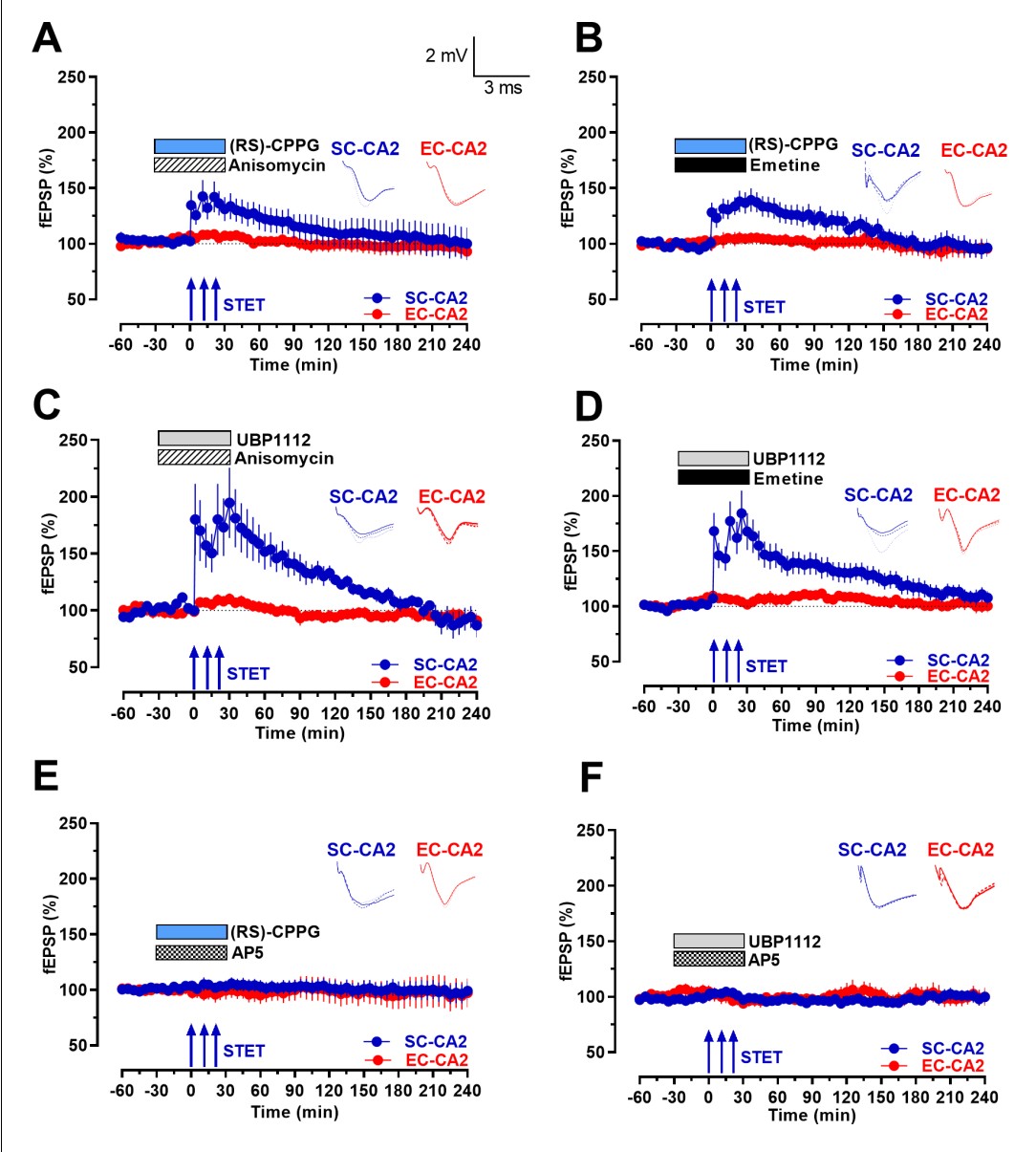

**Figure 4.** Group III mGluR inhibition-mediated activity-dependent late-LTP at Schaffer collaterals to CA2 is protein synthesis- and NMDAR-dependent. (A-B) Co-application of protein synthesis inhibitors anisomycin (25 µM; n = 7; A) or emetine (20 µM; n = 9; B) with (RS)-CPPG for 1 hr prevented late-LTP at SC-CA2 (blue circles) after STET was delivered to SC-CA2. EC-CA2 inputs exhibited stable baseline potentials throughout (red circles). (C–D) Co-application of protein synthesis inhibitors anisomycin (25 µM; n = 6; C) or emetine (20 µM; n = 7; D) with UBP1112 prevented late-LTP at SC-CA2 (blue circles) after STET was delivered to SC-CA2. EC-CA2 inputs exhibited stable baseline potentials throughout (red circles). (E–F) Inhibition of NMDA receptors by co-application of D-AP5 (50 µM) with (RS)-CPPG (n = 10; E) or UBP1112 (n = 6; F) for 1 hr completely abolished LTP induction at SC-CA2 after STET (blue circles). Stable baseline potentials were recorded in EC-CA2 synaptic input (red circles). Horizontal bars indicate drug application period. Representative fEPSP traces 30 min before (closed line), 60 min after (dotted line), and 240 min after (hatched line) STET are depicted. Calibration bars for fEPSP traces in all panels are 2 mV/3 ms. Arrows indicate the time points of STET. 'n' represents number of slices.

potentiation after STET was maintained for 115 min (*Figure 4B*, Wilcoxon test; p=0.028) and up to 165 min (*Figure 4D*, Wilcoxon test; p=0.028), after this time period, the fEPSP gradually decayed back to baseline within 4 hr (p>0.05).

Following a similar experimental design, in the next series of experiments, we co-applied NMDA receptor antagonist D-AP5 (50 µM) with group III mGluR antagonists (RS)-CPPG (*Figure 4E*) or UBP1112 (*Figure 4F*). LTP induction in the SC-CA2 pathway was completely abolished in the presence of D-AP5 (Wilcoxon test; p>0.05 at any given time point) in both cases, thereby indicating that

NMDA receptor activity is crucial for the early induction phase of group III mGluR inhibition-mediated LTP. Stable baseline potentials were recorded in the EC-CA2 synaptic input in all cases (*Figure 4A–F*; red circles; Wilcoxon test; p>0.05).

## Group III mGluR inhibition-mediated plasticity expresses synaptic tagging/capture in the hippocampal area CA2

The synaptic tagging and capture (STC) model, proposed by Frey and Morris in 1997, is a comprehensive model that portrays how late-associativity occurs at the level of synapses in a time-dependent and input-specific manner (*Frey and Morris, 1997*). According to the STC hypothesis, an LTP-like event can induce short-lasting 'tags' that mark the potentiated synapses. It is noteworthy that 'tags' can be set by both weak and strong synaptic events (in this case, weak and strong tetanization to induce early- and late-LTP, respectively), and can eventually capture plasticity-related proteins (PRPs) that are synthesized de novo in response to strong synaptic activity that is sufficient to induce late-LTP. We performed experiments within the STC framework to determine whether late-LTP induction in the presence of group III mGluR antagonists, (RS)-CPPG and UBP1112, can lead to the expression of PRPs that can contribute to the potentiation of 'tagged synapses' and transform an early-LTP in the other synaptic input to a late-LTP ('strong before weak' paradigm (SBW)). After recording a stable baseline for 30 min, a STET was delivered to the SC-CA2 synaptic input in the presence of the group III mGluR antagonists as per the previously mentioned experimental design. A statistically significant potentiation was observed in the SC-CA2 synaptic input immediately after the induction of late-LTP by STET (*Figure 5A & C*, blue circles; statistically significant potentiation at 30 min, Wilcoxon test; p=0.012 and 0.018 in *Figure 5A & C*, respectively) and maintained throughout the experiment (p=0.012 and 0.018 at 240 min in *Figure 5A & C*, respectively). Application of the antagonists was continued for another 30 min and a weak tetanization (WTET) was induced in the EC-CA2 synaptic input 1 hr after the induction of STET in the SC-CA2 synaptic input. Late-LTP was observed in the weakly tetanized EC-CA2 synaptic input in both cases (*Figure 5A & C*; red circles; Wilcoxon test; p=0.012 and p=0.018 at 240 min in *Figure 5A & C* respectively). In contrast, for the same experiment without STET in the SC-CA2 synaptic input, the baseline potentials in SC-CA2 remained stable (*Figure 5B & D*; blue circles; p>0.05). Thirty minutes after the wash out of (RS)-CPPG and UBP1112, WTET was applied in EC-CA2, which resulted in an early-LTP that returned to baseline levels by 125 min and 140 min respectively (*Figure 5B & D*; red circles; Wilcoxon test; p=0.4961 and 0.9375 at 240 min in *Figure 5B & D* respectively). Thus, inhibition of group III mGluRs not only promotes and maintains late-LTP in the SC-CA2 synaptic input but also supports STC-like processes.

## Inhibition of group III mGluRs leads to phosphorylation of ERK and downregulation of STEP

The ERK-MAPK signaling pathway plays a vital role in learning and memory. For instance, the activation of mitogen-activated protein kinases ERK1/2 can lead to the formation of new dendritic spines (*Peng et al., 2010*). Moreover, ERK1/2, through phosphorylating nuclear transcription and translation factors, is critical for the induction and maintenance of LTP (*English and Sweatt, 1996*). Since group III mGluRs are known to inhibit the cAMP/PKA system, which, in turn, activates ERK signaling (*Impey et al., 1998*; *Ni et al., 1998*), it is likely that these mGluRs also inhibit the ERK/MAPK signaling pathway. Thus, to explore the molecular mechanisms supporting the group III mGluR inhibition-mediated plasticity and associativity in CA2, we quantified total and phosphorylated-ERK1/2 proteins in area CA2 isolated from hippocampal slices treated with (RS)-CPPG or UBP1112. Our western blot data showed a significant increase in the phosphorylated-ERK1/2 levels in the antagonist-treated slices in comparison with the control slices (*Figure 6A & C*, one-way ANOVA, p=0.04) with no observable changes in total ERK1/2 levels (*Figure 6A & B*, one-way ANOVA, p>0.05).

The striatal-enriched protein tyrosine phosphatase (STEP) is enriched in pyramidal cells in hippocampal area CA2 (*Boulanger et al., 1995*; *Dudek et al., 2016*). Moreover, STEP is a regulator of ERK1/2 activity and STEP knockout mice have been previously reported to show increased phosphorylated-ERK1/2 levels in the hippocampal CA2 region (*Venkitaramani et al., 2009*). As we saw an increase in phosphorylated-ERK1/2 protein levels in the CA2 region of group III mGluR antagonist treated slices compared to control untreated slices, we wanted to probe if there is a difference in

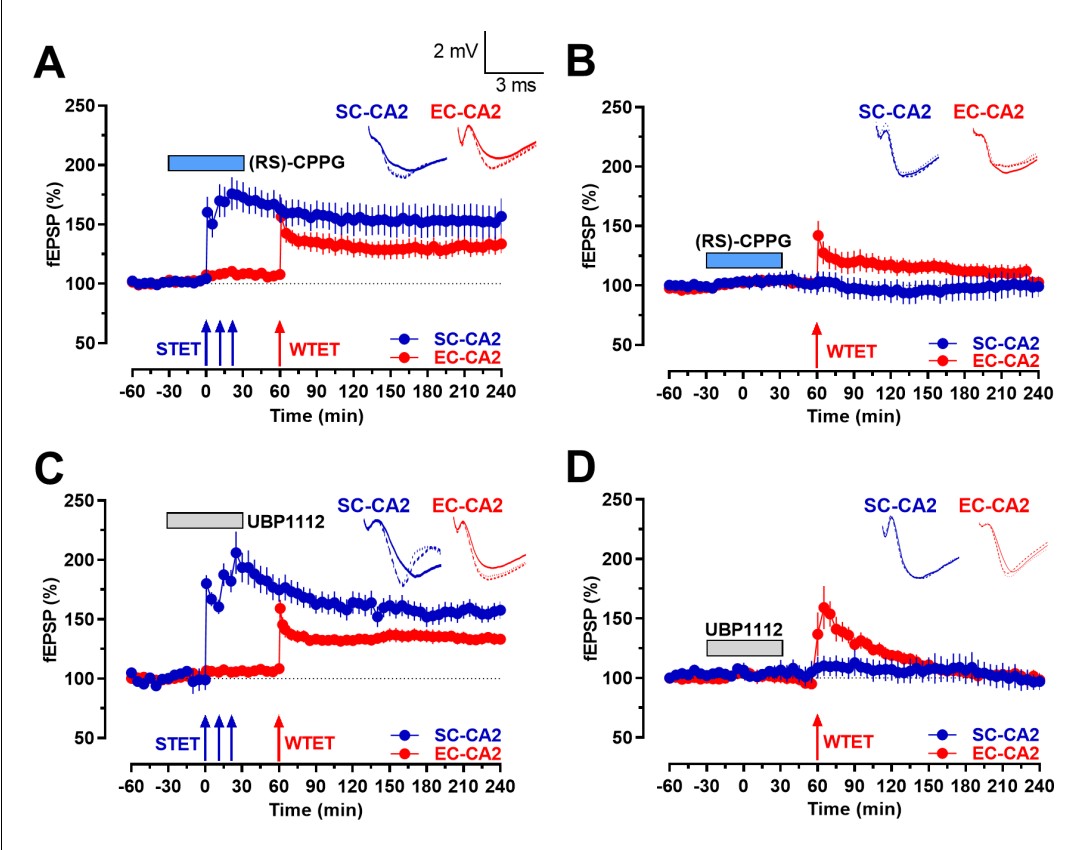

**Figure 5.** Group III mGluR inhibition-mediated activity-dependent late-LTP at Schaffer collaterals to CA2 synapses allows early-LTP to late-LTP transformation at entorhinal cortical synapses to CA2 through synaptic tagging and capture. (**A**) STET at SC-CA2 in the presence of (RS)-CPPG prior to WTET at EC-CA2 transforms early-LTP at EC-CA2 to late-LTP. After recording a stable baseline for 30 min, (RS)-CPPG was bath applied for 1 hr. 30 min into (RS)-CPPG application, STET was delivered to the SC-CA2 synaptic input in the presence of (RS)-CPPG, resulting in late-LTP lasting 4 hr at SC-CA2 (blue circles). WTET was induced at the EC-CA2 synaptic input 1 hr after the induction of STET at SC-CA2. EC-CA2 potentiation in response to WTET was transformed to late-LTP lasting 180 min (red circles; n = 8). (**B**) Experiment shows expression of early-LTP at EC-CA2 after WTET was delivered at the EC-CA2 synaptic input 30 min after (RS)-CPPG application. (RS)-CPPG was applied in total for 1 hr and washed out for 30 min before the delivery of WTET at EC-CA2. The potentiation at EC-CA2 decayed to baseline levels 65 min post-delivery of WTET (red circles; n = 6). Stable baseline potentials were recorded at SC-CA2 (blue circles). (**C**) Similar to A, STET at SC-CA2 (blue circles) in the presence of UBP1112 prior to WTET at EC-CA2 transformed early-LTP at EC-CA2 to late-LTP lasting 180 min (red circles; n = 8). The experimental design was similar to that in B, with the exception that UBP1112 was applied in place of (RS)-CPPG. (**D**) Similar to B, experiment shows expression of early-LTP at EC-CA2 after WTET was delivered at EC-CA2 synaptic input 30 min after UBP1112 application. UBP1112 was applied for 1 hr in total and washed out for 30 min before the delivery of WTET at EC-CA2. The potentiation at EC-CA2 decayed to baseline levels by 140 min (red circles; n = 7). Horizontal bars indicate drug application period. Representative fEPSP traces 30 min before (closed line), 60 min after (dotted line), and 240 min after (hatched line) STET/WTET are depicted. Calibration bars for fEPSP traces in all panels are 2 mV/3 ms. Arrows indicate the time points of WTET or STET. 'n' represents number of slices.

the levels of STEP protein as well in the CA2 region between antagonist-treated and untreated hippocampal slices. Consistent with this line of thought, we observed a significant decrease in the expression levels of STEP protein in the hippocampal area CA2 of slices treated with group III mGluR antagonists (RS)-CPPG and UBP1112 in comparison with untreated hippocampal slices (*Figure 6D & E*, one-way ANOVA, p=0.003).

## Group III mGluR inhibition-mediated plasticity in the hippocampal area CA2 requires MEK activity

Our data showing that group III mGluR antagonists can regulate STEP and ERK expression prompted us to perform a series of electrophysiology experiments to investigate whether the inhibition of ERK1/2 could influence the late-LTP expressed upon the inhibition of group III mGluRs. To achieve that, we used specific inhibitors of the kinases upstream of ERK1/2, MEK1 and MEK2. These

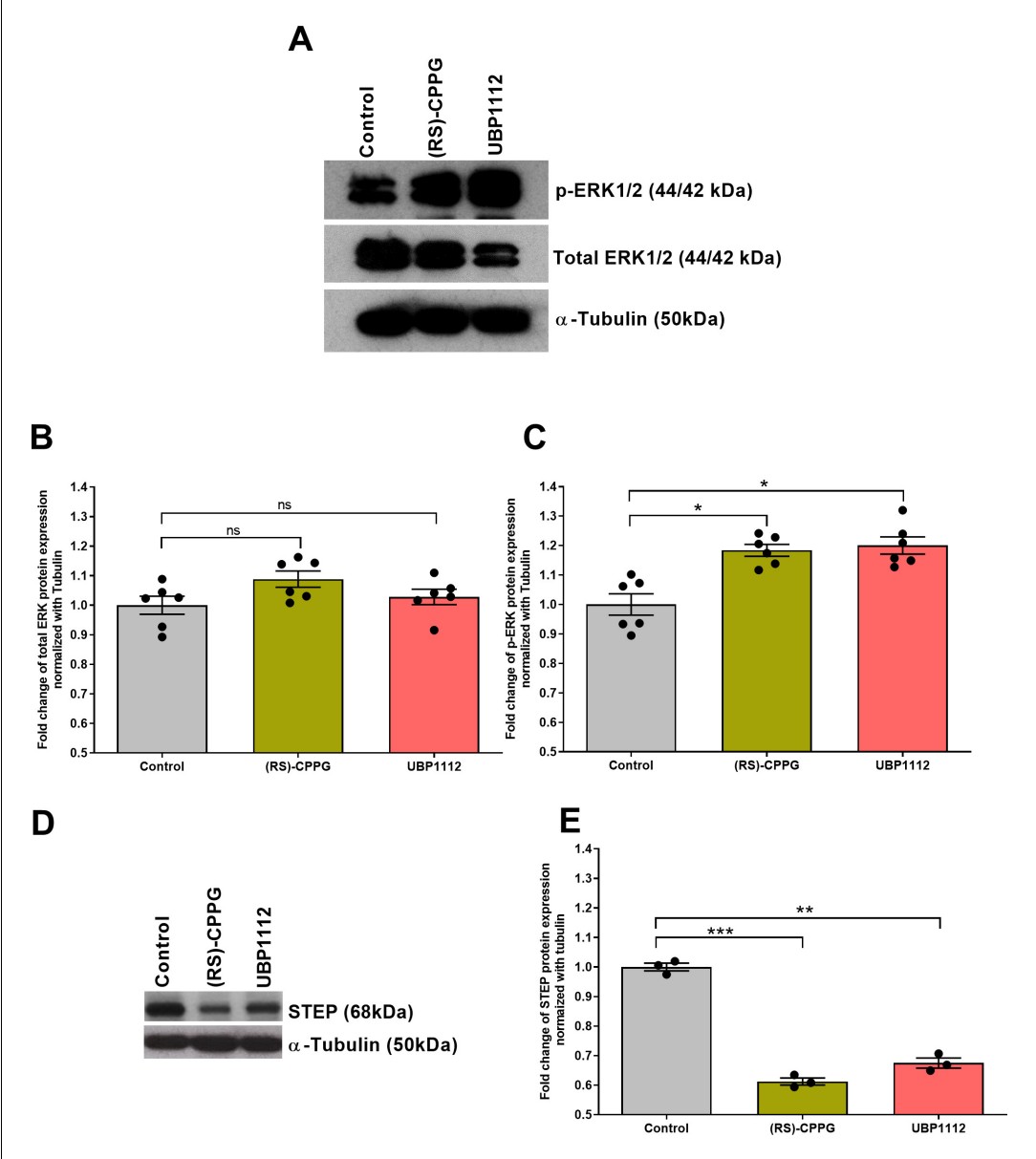

**Figure 6.** Group III mGluR antagonist treatment increases phosphorylation of ERK1/2 and reduces STEP protein expression in hippocampal area CA2. (A-C) Group III mGluR antagonist treatment increases phosphorylation levels of ERK in hippocampal area CA2. (**A**) Total ERK (44/42 kDa), p-ERK (44/42 kDa), and α-Tubulin (50 kDa) immunoreactive bands are shown. (**B**) The expression levels of total ERK also showed an increase after treatment with (RS)-CPPG and UBP1112, however, the data are not statistically significant (one-way ANOVA, 12 slices each from six different biological samples, n = 6). (**C**) Western blot analysis showed a significant increase in the phosphorylation levels of ERK after treatment with (RS)-CPPG or UBP1112 when compared to the control group. Significant differences between the groups control vs (RS)-CPPG or UBP1112 are indicated by * *P* < 0.05 (one-way ANOVA, 12 slices each from six different biological samples, n = 6). In both (**B** and **C**), the data were normalized to their respective tubulin levels. (**D**) STEP (68 kDa) and α-Tubulin (50 kDa) immunoreactive bands are shown. (**E**) Western blot analysis showed a significant reduction in STEP protein expression levels in (RS)-CPPG- or UBP1112-treated groups when compared to control group in area CA2. The data were normalized to their respective tubulin levels. Significant differences between the groups control vs (RS)-CPPG or UBP1112 are indicated by **: *P* < 0.01, ***: *P* < 0.001 (one-way ANOVA, 12 slices each from three different biological samples, n = 3). 'n' represents number of animals.

MEK inhibitors, U0126 or PD98059, were co-applied with group III mGluR antagonists, (RS)-CPPG or UBP1112, for a total time period of 1 hr. A statistically significant potentiation was observed in the SC-CA2 synaptic input immediately after the induction of STET during the co-application of (RS)-CPPG with U0126 (*Figure 7A*; blue circles; Wilcoxon test; p=0.028 at 30 min after STET) or PD98059

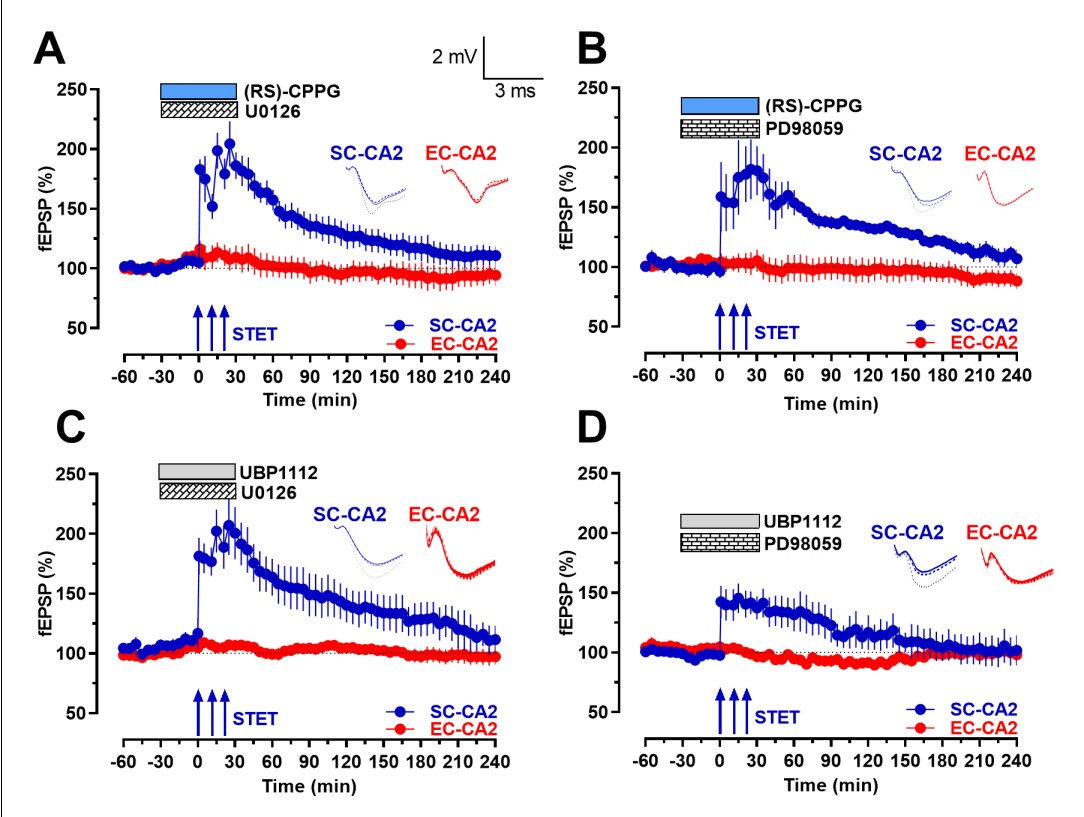

**Figure 7.** Inhibition of upstream kinases of ERK1/2 prevents group III mGluR inhibition mediated activity-dependent late-LTP at Schaffer collateral inputs to CA2. (A-B) Co-application of specific inhibitors of the upstream kinases of ERK1/2, U0126 (1 µM; n = 6; A) or PD98059 (1 µM; n = 6; B), with (RS)-CPPG abolished late-LTP after STET at SC-CA2 synaptic input (blue circles; A B). U0126 or PD98059 were co-applied with (RS)-CPPG for a total period of 1 hr and STET was delivered at SC-CA2 30 min into drug application. The potentiation in SC-CA2 after the induction of STET gradually decayed back to baseline potentials (blue circles). Stable baseline potentials were recorded in EC-CA2 synaptic input throughout (red circles). (C–D) Similar to A and B, co-application of U0126 (1 µM; n = 6; C) or PD98059 (1 µM; n = 6; D) with UBP1112 abolished late-LTP after STET at SC-CA2 synaptic input (blue circles; C D) and fEPSPs at SC-CA2 gradually decayed back to baseline potentials in both cases. The experimental design was similar to that in A and B, with the exception that UBP1112 was applied in place of (RS)-CPPG. Stable baseline potentials were recorded in EC-CA2 synaptic input throughout (red circles). Horizontal bars indicate drug application period. Representative fEPSP traces 30 min before (closed line), 60 min after (dotted line), and 240 min after (hatched line) STET are depicted. Calibration bars for fEPSP traces in all panels are 2 mV/3 ms. Arrows indicate the time points of STET. 'n' represents number of slices.

(*Figure 7B*; blue circles; Wilcoxon test; p=0.018 at 30 min after STET) but gradually decayed back to the baseline and was not significantly different from baseline from 90 min (*Figure 7A*, Wilcoxon test; p=0.075) and from 160 min (*Figure 7B*; Wilcoxon test; p=0.063) onwards. Similarly, in the case of UBP1112, the potentiation was statistically significantly immediately after the induction of STET during the co-application with U0126 (*Figure 7C*; blue circles; Wilcoxon test; p=0.028 at 30 min after STET) or PD98059 (*Figure 7D*; blue circles; Wilcoxon test; p=0.028, at 30 min after STET) but was no longer significant from 90 min (*Figure 7C*; Wilcoxon test; p=0.074) and 95 min (*Figure 7D*; Wilcoxon test; p=0.075) onwards. Stable baseline potentials were recorded in the EC-CA2 synaptic input in all cases (*Figure 7A–D*; red circles; Wilcoxon test; p>0.05). Thus, our data show strong evidence that inhibition of group III mGluRs helps maintain long-lasting LTP in the hippocampal area CA2 via the ERK/MAPK signaling pathway.

## Discussion

Although we now know that hippocampal CA2 plays a crucial role in social memory formation (*Hitti and Siegelbaum, 2014*; *Tzakis and Holahan, 2019*), we still do not fully understand how the CA2 region's unique molecular and cellular properties contribute to its function at the circuit and

behavioural level. Of particular interest, Schaffer collateral synapses that project onto CA2 (SC-CA2) do not express canonical activity-dependent LTP, unlike SC-CA1 synapses (*Zhao et al., 2007*; *Caruana et al., 2012*). Reasons for this include expression of particular genes, such as Regulator of G protein signalling 14 (RGS14) (*Lee et al., 2010*; *Evans et al., 2018*), and extracellular matrix components (*Carstens et al., 2016*), as well as higher calcium buffering and calcium extrusion in CA2 spines (*Simons et al., 2009*). In addition, area CA2 has a higher density of interneurons as compared to area CA1 (*Botcher et al., 2014*). Activity-dependent plasticity of CA2 interneurons, namely long-term depression at inhibitory synapses (iLTD), has been shown to increase the excitatory drive between CA3 and CA2, allowing SC inputs to drive action potential firing in CA2 pyramidal neurons (PNs), thus providing a mechanism whereby CA2 PNs can be engaged by CA3 (*Nasrallah et al., 2015*). Nevertheless, what contributes to the plasticity-resistant property at only the proximal Schaffer collateral synapses of CA2 neurons is not fully understood. Indeed, SC-CA2 synapses exhibit synaptic potentiation under certain conditions (*Benoy et al., 2018*; *Carstens and Dudek, 2019*). For example, a previous study from our lab has demonstrated that bath application of the neuropeptide substance P can elicit NMDAR-dependent lasting synaptic potentiation at both SC and EC synapses at CA2 (*Dasgupta et al., 2017*). In this study, we investigated the role of group III mGluRs in synaptic plasticity in CA2 using specific pharmacological inhibitors (RS)-CPPG and UBP1112. The most striking result is that an activity-dependent Hebbian LTP, that is NMDAR- and protein synthesis-dependent, becomes inducible at SC-CA2 synapses upon inhibition of group III mGluRs. The results from this study also suggest that group III mGluR inhibition-mediated LTP at SC-CA2 synapses is mechanistically postsynaptic and inhibition of these receptors decreases the threshold for inducing LTP only in SC-CA2 synapses. Furthermore, inhibition of group III mGluRs facilitates associative plasticity through synaptic tagging and capture at the entorhinal cortical and Schaffer collateral synaptic inputs to CA2 neurons.

This observed modulation of synaptic plasticity in the CA2 region by group III mGluRs is dependent on the ERK/MAPK pathway, which is important for the maintenance of LTP in other hippocampal regions (*English and Sweatt, 1996*; *Rosenblum et al., 2002*). Our result suggests that STEP is downregulated when group III mGluRs are inhibited during the induction of LTP. As the protein-tyrosine phosphatase STEP is a known inactivator of ERK1/2 (*Pulido et al., 1998*), this reduction in STEP could have contributed to the increase in ERK1/2 phosphorylation observed after group III mGluR inhibition-mediated LTP. In turn, this increased activation of ERK could have led to the activation of downstream mechanisms that sustain synaptic potentiation, such as de novo protein synthesis. Indeed, studies have shown that the loss of STEP leads to enhanced hippocampal LTP (*Zhang et al., 2010*) and memory (*Venkitaramani et al., 2011*) in mice. Moreover, STEP interacts with NMDA receptors and has been shown to be a critical regulator of NMDA receptor trafficking and function as the reduction in STEP expression by RNA interference results in increased NMDAR function and surface expression of NR1, NR2A and NR2B receptor subunits (*Braithwaite et al., 2006*). These are suggestive of the possible functional implications of the observed reduction in STEP levels in hippocampal area CA2 upon group III mGluR inhibition.

The notion that group III mGluR inhibition facilitates induction and maintenance of potentiation through enhancing ERK signaling pathway and the resulting synthesis of plasticity-related proteins (PRPs) is supported by our results that MEK inhibitors and translation inhibitors abrogated mGluR inhibition-mediated LTP. This elevation of PRPs could also underlie our observation of the expression of synaptic tagging/capture in CA2 in the presence of group III mGluR antagonists. Frey and Morris demonstrated in their seminal paper that a transient form of LTP (early-LTP), induced by a weak stimulus, can be converted to a more persistent form of LTP (late-LTP), induced by a strong stimulus, if the two stimuli are applied on different synapses of the same neuron within a time frame of 60 min (*Frey and Morris, 1997*; *Redondo and Morris, 2011*). The 'tag', formed as a result of weak stimulation, can capture the newly synthesized PRPs, provided by the strong stimulation, in order to sustain the LTP (*Frey and Morris, 1997*; *Redondo and Morris, 2011*). Typically, SC-CA2 synapses do not engage in STC, but we observed STC-like processes in the presence of (RS)-CPPG and UBP1112. This substantiates the notion that inhibition of group III mGluRs not only provides a conducive environment for LTP induction and maintenance in CA2 neurons but also promotes PRP synthesis to a level high enough to help associate weak information with strong information in a time-dependent manner.

It is to be noted that the high relative expression of mGluR4 in the CA2 region appears to be restricted to the rat hippocampus (*Ohishi et al., 1995*; *Phillips et al., 1997*). Similar enrichment of mGluR4 in CA2 region has not been reported from expression studies in the mouse hippocampus (*Farris et al., 2019*). However, functionally, both species display similar plasticity resistance at CA2 synapses (*Zhao et al., 2007*; *Dasgupta et al., 2017*). This suggests possible mechanistic differences in the regulation of synaptic plasticity in the CA2 region across species.

In conclusion, our study suggests that group III mGluRs gate SC-CA2 synaptic plasticity by suppressing ERK signaling and maintaining an elevated expression of STEP. Blocking group III mGluR activity during high-frequency stimulation can facilitate the induction of LTP, as well as late-associative plasticity, in a protein synthesis- and NMDAR-dependent manner. However, the implications of group III mGluR-mediated regulation of SC-CA2 plasticity and associativity on social memory performance remain to be determined.

# Materials and methods

## Key resources table

| Reagent type (species) or resource | Designation | Source or reference | Identifiers | Additional information |
|---|---|---|---|---|
| Chemical compound, drug | (RS)-CPPG | Santa Cruz Biotechnology | Cat. #: sc-203448 | 1 µM final concentration |
| Chemical compound, drug | UBP1112 | Santa Cruz Biotechnology | Cat. #: sc-204368 | 15 µM final concentration |
| Chemical compound, drug | Emetine dihydrochloride | Sigma-Aldrich | Cat. #: E2375 | 20 µM final concentration |
| Chemical compound, drug | D-AP5 | Tocris | Cat. #: 0106 | 50 µM final concentration |
| Chemical compound, drug | Anisomycin | Tocris | Cat. #: 1290 | 25 µM final concentration |
| Chemical compound, drug | U0126 | Promega | Cat. #: V1121 | 1 µM final concentration |
| Chemical compound, drug | PD98059 | Cell Signaling Technology | Cat. #: 9900L | 1 µM final concentration |
| Chemical compound, drug | Picrotoxin | Sigma-Aldrich | Cat. #: P1675 | 100 µM final concentration |
| Antibody | rabbit polyclonal anti-P42/44 MAPK or ERK1/2 | Cell Signaling Technology | Cat. #: 9102S | WB (1:500) |
| Antibody | rabbit monoclonal anti-phospho-P42/44 MAPK or p-ERK1/2 | Cell Signaling Technology | Cat. #: 4370S | WB (1:500) |
| Antibody | mouse monoclonal IgG$_{2b}$ anti-STEP | Santa Cruz Biotechnology | Cat. #: sc-23892 | WB (1:500) |
| Antibody | mouse monoclonal anti-tubulin antibody | Sigma-Aldrich | Cat. #: T9026 | WB (1:30000) |
| Antibody | anti-rabbit IgG, HRP-linked secondary antibody | Cell Signaling Technology | Cat. #: 7074 | WB (1:3000) |
| Commercial assay or kit | Supersignal West Pico | Thermo Scientific | Cat. #: 34580 | |
| Commercial assay or kit | RNeasy Mini kit | Qiagen | Cat. #: 74106 | |
| Commercial assay or kit | GoScript Reverse Transcription System | Promega | Cat. #: A5000 | |
| Commercial assay or kit | Taqman universal PCR master mix | Thermo Fisher Scientific | Cat. #: 4304437 | |
| Sequence-based reagent | TaqMan gene expression assay mGluR4/Grm4 | Thermo Fisher Scientific | Lot. #: 1530835 Assay ID: Rn 01428450 | |

*Continued on next page*

*Continued*

| Reagent type (species) or resource | Designation | Source or reference | Identifiers | Additional information |
|---|---|---|---|---|
| Sequence-based reagent | TaqMan gene expression assay mGluR6/Grm6 | Thermo Fisher Scientific | Lot. #: 1534985 Assay ID: Rn 00709483 | |
| Sequence-based reagent | TaqMan gene expression assay mGluR7/Grm7 | Thermo Fisher Scientific | Lot. #: 1594998 Assay ID: Rn 00667503 | |
| Sequence-based reagent | TaqMan gene expression assay mGluR8/Grm8 | Thermo Fisher Scientific | Lot. #: 1407050 Assay ID: Rn 00573505 | |

## Preparation of hippocampal slices

A total of 154 acute hippocampal slices prepared from 97 male Wistar rats (5–7 weeks old) were used for the study. All animal procedures were approved by the Institutional Animal Care and Use Committee (IACUC- protocol approval number R16-0135) of the National University of Singapore. The rats were anesthetized using $CO_2$ and decapitated to take out the brains quickly, which were then placed in 4°C artificial cerebrospinal fluid (aCSF) containing the following (in mM): 124 NaCl, 2.5 KCl, 2 $MgCl_2$, 2 $CaCl_2$, 1.25 $NaH_2PO_4$, 26 $NaHCO_3$, 17 D-Glucose equilibrated with 95% $O_2$ and 5% $CO_2$ (carbogen). Transverse hippocampal slices (400 μm in thickness) were prepared from the right hippocampus with a manual tissue chopper. For field potential recordings, the slices were then transferred onto a nylon net in an interface chamber (Scientific Systems Design, Ontario, Canada) and incubated at 32°C. An aCSF flow of 1 mL/min and carbogen consumption of 16 L/h were maintained throughout. The whole process from anaesthetization to the transfer of slices to the chamber was fast and did not exceed the average duration of 5 min. The slices were incubated for at least 2–3 hr before starting the experiments.

## Field potential recordings

For most of the experiments, monopolar lacquer-coated stainless-steel electrodes were positioned in the hippocampal area CA2. The stimulating electrodes were located in the Schaffer collaterals (CA3 → CA2) and cortical fibers in the *Stratum lacunosum-moleculare* (Entorhinal cortex → CA2) and field excitatory postsynaptic potentials (fEPSPs) were recorded from the distal dendritic region of the CA2 neurons (*Figure 1A*). The independence of the two synaptic pathways was ensured by a cross-input paired-pulse facilitation (PPF) protocol with an interpulse interval of 50 ms similar to that of our earlier report (*Dasgupta et al., 2017*). Similar to our previous study, the absence of PPF indicated that the two stimulation electrodes activate independent SC and EC synaptic inputs to CA2 neurons (*Dasgupta et al., 2017*). The stimulation strength for the experiments was determined by an input-output curve (stimulus intensity vs. field EPSP slope) and set to obtain an fEPSP slope of 40% of the maximum slope value. A custom-made software, PWIN (Leibniz Institute for Neurobiology, Magdeburg, Germany, can be provided upon request), was used to regulate stimulation and to record and monitor fEPSP signals online (*Shetty et al., 2015*).

A stable baseline was recorded for at least 30 min prior to the application of pharmacological substances and electrical LTP induction. Four 0.2 Hz biphasic constant current pulses were used for baseline and post-induction recording. For early long-term potentiation (early-LTP) induction, a weak induction protocol (WTET), consisting of a single high frequency stimulation (100 Hz, 21 pulses, single burst, 0.2 ms pulse duration) was used, whereas for late long-term potentiation (late-LTP) induction, a strong tetanization protocol (STET) involving repeated high-frequency stimulation protocol (three trains of 100 Hz, 100 pulses, single burst, 0.2 ms pulse duration) was used with an inter-train interval of ten minutes.

Similar to our previous study (*Dasgupta et al., 2017*), bath medium used in all field potential recordings in this study did not contain GABA receptor antagonists and thus had inhibitory transmission intact.

## Whole-cell Voltage-Clamp recordings

Slices were prepared according to the method described in our earlier reports (*Krishna et al., 2016*; *Dasgupta et al., 2017*). All recordings were made in the whole-cell voltage-clamp configuration at a holding potential of −70 mV. A triple patch-clamp EPC10 patch-clamp amplifier and the software

Patchmaster (HEKA Electronics, Lambrecht, Germany) were used for data acquisition as described previously (*Krishna et al., 2016*; *Dasgupta et al., 2017*). In addition, a CED 1401 analog-to-digital converter (Cambridge Electronic Design) and a custom-made software program, PWIN, were used to regulate stimulation and to record evoked excitatory postsynaptic currents (evoked EPSCs). Evoked EPSCs were recorded from the soma of visually identified pyramidal neurons located within the CA2 region of the hippocampus. Patch pipettes (4–6 MΩ) were filled with internal solution containing (in mM): 135 CsMeSO$_3$, 8 NaCl, 10 HEPES, 0.25 EGTA, 2 Mg$_2$ATP, 0.3 Na$_3$GTP, 0.1 spermine, seven phospho-creatine, and 5 QX-314 (pH 7.3). The GABA$_A$ receptor antagonist, picrotoxin (100 µM), was added to the bath medium in all recordings. Recordings were considered stable when the membrane resistance, membrane capacitance, or the holding current did not change more than 20%.

EPSCs were evoked by the stimulation of Schaffer-collateral fibers in the *stratum radiatum* of CA2 using tungsten stimulating electrode (AM systems, WA, USA). Schaffer collateral projections were stimulated at 0.05 Hz by 200 µs voltage pulses generated by an isolated pulse stimulator (Model 2100, AM systems). The stimulation strength (2–5 V) was adjusted to evoke EPSCs with an amplitude of approximately 50 pA. Baseline synaptic currents were recorded for at least 10 min prior to LTP induction. LTP was induced within 10 min after whole-cell break-in by a one second 100 Hz stimulation, synchronized with a membrane depolarization to 0 mV for two seconds through the patch-pipette (paired HFS (dep. to 0 mV + 100 Hz/s)). The slope of EPSCs was measured to analyze the change in synaptic transmission before and after induction of LTP. The EPSC slope values were normalized to the first 10 min of EPSC recording.

## Pharmacology

(RS)-CPPG (Santa Cruz # sc-203448) is a potent group III/II antagonist with 20-fold more selectivity for group III over group II (*Schoepp et al., 1999*). It was stored at −20˚C as 1 mM stock in deionized water. Before application, the stock solution was diluted to a final concentration of 1 µM in aCSF and applied for a total of 60 min (30 min before and after the induction of STET) unless otherwise specified. Another more selective group III mGluR inhibitor UBP1112 (Santa Cruz # sc-204368) was stored at −20˚C as 75 mM stock in 1 N NaOH (*Miller et al., 2003*). The stock was diluted to a final concentration of 15 µM and applied for a total of 60 min (30 min before and after the induction of STET) unless otherwise specified. Emetine dihydrochloride (Sigma-Aldrich #E2375), a protein synthesis inhibitor, was stored at −20˚C as 20 mM stock in deionized water. The stock was diluted to a final working solution of 20 µM in aCSF and co-applied with (RS)-CPPG/UBP1112 for 1 hr. Another protein synthesis inhibitor Anisomycin (Tocris #1290) was stored at −20˚C as a 25 mM stock in DMSO. It was diluted to 25 µM in aCSF and co-applied with (RS)-CPPG/UBP1112 for 1 hr. A 50 mM stock of D-AP5 (Tocris #0106), a competitive NMDA receptor antagonist, was prepared in deionized water and stored at −20˚C. A working solution of 50 µM was made in aCSF and co-applied for 1 hr with (RS)-CPPG/UBP1112. The MEK inhibitor U0126 (#V1121, Promega, Madison, Wisconsin) was stored as 20 mM stock in DMSO. The other MEK inhibitor PD98059 (#9900L, Cell Signaling Technology, Beverly, Massachusetts) was stored as 50 mM stock in DMSO at −20˚C. Both the MEK inhibitors were used at a concentration of 1 µM (dissolved in 0.1% DMSO) (*Davies et al., 2000*; *Navakkode et al., 2005*) just before co-application with (RS)-CPPG/UBP1112. The GABA$_A$ receptor antagonist picrotoxin (Sigma-Aldrich #P1675) was stored at −20˚C as a stock solution of 500 mM in DMSO. The antagonist was diluted in aCSF to a final concentration of 100 µM. In the case of pharmacological substances prepared in DMSO, the DMSO concentration in the final bath-application solution was kept below 0.1%, a concentration that has no effect on basal synaptic responses (*Navakkode et al., 2005*).

## Statistics

The average values of the slope function of field excitatory postsynaptic potentials (fEPSPs) and excitatory postsynaptic currents (EPSCs) were analyzed by Wilcoxon signed-rank test (henceforth 'Wilcoxon test') when compared within the same group (before and after induction of synaptic plasticity). The Mann-Whitney U test was applied when the values were compared between groups. On occasion, Kruskal-Wallis test was used to compare mean fEPSP values across three or more groups. One-way ANOVA at the $p < 0.05$ significance level was used to analyze the western blot and qRT

PCR results. The nonparametric test was used because the analyses of prolonged recordings do not allow the use of parametric tests. Furthermore, the sample sizes did not always guarantee a Gaussian normal distribution of the data per series. GraphPad Prism version 8.0.1 for Windows (GraphPad Software, San Diego, California, USA, www.graphpad.com) was used to plot graphs and to perform statistical tests. Detailed descriptions of statistical analysis of each experiment are provided in the results section.

## Western blot assay

Rat hippocampal slices (5–7 weeks old, three biological samples were used for each group, n = 3; 'n' represents number of animals) from three groups: control, (RS)-CPPG and UBP1112, were collected after electrophysiology recordings, snap frozen in liquid nitrogen and stored at −80°C. CA2 regions were isolated carefully from the frozen hippocampal slices similar to our previous report (*Dasgupta et al., 2017*). Total protein from the hippocampal CA2 regions was extracted using the T-PER Tissue Protein Extraction Kit (#78510, Thermo Fisher Scientific Inc, USA), and the HaltTM Protease Inhibitor Cocktail Kit (#78410, Thermo Fisher Scientific Inc, Rockford, IL, USA). A protein assay kit was used to quantify the protein level in the samples (#500–0007, Bio-Rad, Hercules, CA, USA). 20 mg of protein extracts were separated on 10% SDS-polyacrylamide gels and transferred to polyvinylidene difluoride transfer membranes. The membranes were blocked with 5% non-fat dry milk and incubated with primary antibodies overnight at 4°C. The primary antibodies used were rabbit anti-P42/44 MAPK or ERK1/2 (1:500; #: 9102S, Cell Signaling Technology, USA) and rabbit anti-p-P42/44 MAPK or p-ERK1/2 (1:500;: 4370S, Cell Signaling Technology, USA), mouse anti-STEP (1:500; # sc-23892, Santa Cruz Biotechnology) and mouse anti-tubulin monoclonal antibody (# T9026, Sigma-Aldrich, St. Louis, MO, USA). Membranes were incubated with the horseradish peroxidase-conjugated secondary antibody on the following day (# 7074, Cell Signaling Technology, USA) for 1 hr. The immunoproducts were detected using a chemiluminescence detection system according to the manufacturer's instructions (Supersignal West Pico Horseradish Peroxidase Detection Kit, Pierce Biotechnology, IL, USA) and developed on a film. ImageJ software (*Schneider et al., 2012*) was used to quantify the optical density of each protein band. Each lane of protein band density was normalized with its corresponding α-tubulin protein density. Mean protein band density was then calculated using Microsoft Excel (Microsoft Corporation, 2019, https://office.microsoft.com/excel) and statistical analysis was performed using GraphPad Prism version 8.0.1 for Windows (GraphPad Software, San Diego, California, USA, www.graphpad.com).

## RNA extraction and Real-Time quantitative RT-PCR

Rat hippocampal slices (5–7 weeks old, four biological samples were used for each gene expression analysis, n = 4; 'n' represents number of animals) were snap frozen in liquid nitrogen and stored at −80°C. CA2 regions were isolated carefully from the frozen hippocampal slices. Total RNA was extracted from the CA2 hippocampal regions using RNeasy Mini kit according to the manufacturer's instructions (#74106, Qiagen, USA) and quantified using a spectrophotometer (NanoDrop2000, Thermo Scientific). cDNA synthesis was carried out using GoScript Reverse Transcription System (#A5000, Promega). Briefly, 1 µg of RNA was subjected to pre-heating with 2 µl Oligo (dT) at 72°C for 2 min. Reverse transcription was performed at 42°C for 1 hr followed by 95°C for 5 min. Further, StepOne Plus Real-time PCR system (Applied Biosystems) was used to carry out the qRT-PCR with Taqman universal PCR master mix (#4304437, Thermo Scientific) and TaqMan probes specific for group III mGluRs (mGluR4/Grm4, Rn 01428450, Lot. 1530835, mGluR6/Grm6, Rn 00709483, Lot.1534985, mGluR7/Grm7, Rn 00667503, Lot. 1594998, mGluR8/Grm8, Rn 00573505, Lot. 1407050). The qRT-PCR reaction was performed in 96 well plates with an initial denaturation at 95°C for 10 min, followed by 40 amplification cycles, each at 95°C for 15 s, and then at 60°C for 1 min. The target gene expression levels were measured in duplicates and normalized with the internal control GAPDH (Gapdh, Rn01775763_g1, Lot no. 1523580). Fold changes of target genes' expression was calculated according to the $2^{-\Delta\Delta Ct}$ method (*Livak and Schmittgen, 2001*) using Microsoft Excel (Microsoft Corporation, 2019, https://office.microsoft.com/excel) and statistical analysis was performed using GraphPad Prism version 8.0.1 for Windows (GraphPad Software, San Diego, California, USA, www.graphpad.com).

## Acknowledgements

We thank Dr. Serena M Dudek (National Institute of Environmental Health Sciences, National Institutes of Health) for her constructive comments on the manuscript. This work was supported by National Medical Research Council Collaborative Research Grants (NMRC-CBRG-0099–2015 and NMRC/OFIRG/0037/2017), Ministry of Education Academic Research Fund Tier 3 (MOE2017-T3-1-002), National University of Singapore, University Strategic Research Grant (DPRT/944/09/14) and National University of Singapore Yong Loo Lin School of Medicine Aspiration Fund (Grant R-185-000-271-720) to SS. TB was supported by NSFC (31871076). AD was supported by NUS Research Scholarship.

## Additional information

### Funding

| Funder | Grant reference number | Author |
|---|---|---|
| National Medical Research Council | NMRC-CBRG-0099-2015 | Sajikumar Sreedharan |
| National Medical Research Council | NMRC/OFIRG/0037/2017 | Sajikumar Sreedharan |
| Ministry of Education - Singapore | MOE2017-T3-1-002 | Sajikumar Sreedharan |
| National University of Singapore | DPRT/944/09/14 | Sajikumar Sreedharan |
| National University of Singapore | R-185-000-271-720 | Sajikumar Sreedharan |
| National Natural Science Foundation of China-Yunnan Joint Fund | 31871076 | Thomas Behnisch |
| National University of Singapore | Research Scholarship | Amrita Benoy |

The funders had no role in study design, data collection and interpretation, or the decision to submit the work for publication.

### Author contributions

Ananya Dasgupta, Conceptualization, Data curation, Investigation, Writing - original draft; Yu Jia Lim, Data curation, Formal analysis, Investigation; Krishna Kumar, Data curation, Formal analysis, Investigation, Methodology; Nimmi Baby, Ka Lam Karen Pang, Data curation, Formal analysis, Validation, Investigation; Amrita Benoy, Data curation, Validation, Investigation, Writing - original draft; Thomas Behnisch, Conceptualization, Formal analysis, Validation, Writing - review and editing; Sreedharan Sajikumar, Conceptualization, Resources, Data curation, Software, Formal analysis, Supervision, Funding acquisition, Validation, Investigation, Methodology, Writing - original draft, Project administration

### Author ORCIDs

Sreedharan Sajikumar ⓘD https://orcid.org/0000-0002-5761-8982

### Ethics

Animal experimentation: All animal procedures were approved by the Institutional Animal Care and Use Committee (IACUC) of the National University of Singapore.(IACUC- protocol approval number R16-0135).

### Decision letter and Author response

Decision letter https://doi.org/10.7554/eLife.55344.sa1
Author response https://doi.org/10.7554/eLife.55344.sa2

## Additional files

### Supplementary files
• Transparent reporting form

### Data availability
All data generated or analysed during this study are included in the manuscript and supporting files. Source data files have been uploaded in Open Science Framework (http://doi.org/10.17605/OSF.IO/GFAPD).

The following dataset was generated:

| Author(s) | Year | Dataset title | Dataset URL | Database and Identifier |
|---|---|---|---|---|
| Sajikumar S | 2020 | Group III metabotropic glutamate receptors gate long-term potentiation and synaptic tagging/capture in rat hippocampal area CA2 | http://doi.org/10.17605/OSF.IO/GFAPD | Open Science Framework, 10.17605/OSF.IO/GFAPD |

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
