## [Decision Letter]

**Acceptance summary:**

This paper demonstrates that pharmacological inhibition of type III mGluRs is sufficient to allow NMDAR-dependent LTP in the Schaffer collateral to CA2 pathway. That unmasked LTP has all the features that define Hebbian LTP, including the capacity to associatively support long-term LTP in other synapses. The observation that classical Hebbian LTP is actively suppressed by mGluRIII receptors in CA2 is novel and significant, considering the emerging and increasing interest in this hippocampal subfield.

**Decision letter after peer review:**

Thank you for submitting your article "Group III mGluRs gate long-term potentiation and synaptic tagging/capture in rat hippocampal area CA2" for consideration by *eLife*. Your article has been reviewed by two peer reviewers, and the evaluation has been overseen by a Reviewing Editor and Olga Boudker as the Senior Editor. The following individual involved in review of your submission has agreed to reveal their identity: Alfredo Kirkwood (Reviewer #2).

The reviewers have discussed the reviews with one another and the Reviewing Editor has drafted this decision to help you prepare a revised submission.

Summary:

The authors show that pharmachological inhibition of type III mGluRs is sufficient to allow "standard" NMDAR dependent LTP in the Schaffer collateral to CA2 pathway. That unmasked LTP has all the features that define Hebbian LTP, including the capacity to associatively support long-term LTP in other synapses. The observation that classical Hebbian LTP is actively suppresed in CA2 is novel and significant, considering the emerging -and increasing- interest in this hippocampal subfield. This novel finding should be of general interest to researchers studying synaptic plasticity. The experiments were well executed, and the results are convincing. The reviewers suggest strengthening several important points to support the findings:

Essential revisions:

1) One obvious limitation concerns the question of whether the type III mGluRs limit LTP by acting pre- or postsynaptically. These mGluRs are mostly presynaptic whereas the induction and expression of classical LTP is postsynaptic. Thus, a simple scenario would be that the type III mGluRs somehow limit transmission at high frequencies, preventing the induction of LTP with tetanic stimulation. The presynaptic scenario is easy to check. If that were the case, the induction LTP with pairing protocols (low freq. stim. + depolarization to 0mV) should work even in the absence of the antagonists. Whether the antagonist improve high frequency transmission specifically in the SC-CA2 pathway.

2) Within CA2, is the threshold for LTP at entorhinal inputs altered by the group III mGluR antagonist? Compared to other hippocampal areas, is the LTP-enhancing effect of the antagonist restricted to CA2?

---

## [Author Response]

Essential revisions:1) One obvious limitation concerns the question of whether the type III mGluRs limit LTP by acting pre- or postsynaptically. These mGluRs are mostly presynaptic whereas the induction and expression of classical LTP is postsynaptic. Thus, a simple scenario would be that the type III mGluRs somehow limit transmission at high frequencies, preventing the induction of LTP with tetanic stimulation. The presynaptic scenario is easy to check. If that were the case, the induction LTP with pairing protocols (low freq. stim. + depolarization to 0mV) should work even in the absence of the antagonists. Whether the antagonist improve high frequency transmission specifically in the SC-CA2 pathway.

We are very thankful to the reviewers for the insight with this suggestion. Indeed, we have used a similar kind of pairing protocol for the induction of LTP in SCCA2 by providing 1x100Hz stimulation coupled with depolarization to 0 mV (dep. to 0 mV + 100Hz/s). This was not explicitly represented in Figure 3 initially. To avoid further confusion, we have now replaced œHFS representation in Figure 3 with œPaired HFS (dep. to 0 mV + 100Hz/s). Using this protocol, we could not induce long-lasting LTP at SCCA2 synapses (Figure 3B).

To check whether the type III mGluRs limit LTP by acting pre- or postsynaptically, we have now analyzed the LTP induction traces from single cell and field recording experiments from three groups: (1) without type III mGluR inhibitors, (2) with (RS)-CPPG and (3) with UBP1112. In short, we could not find any specific pre-synaptic enhancement in the presence of the drugs. We have now included these findings in the Results section and the data are now presented in the manuscript as Figure 3â€”figure supplement 1 A-F and Figure 3â€”figure supplement 2 A-C.

2) Within CA2, is the threshold for LTP at entorhinal inputs altered by the group III mGluR antagonist? Compared to other hippocampal areas, is the LTP-enhancing effect of the antagonist restricted to CA2?

We are thankful to the reviewers for the insight with this question. We have now tested this idea by inducing an early-LTP (1x100 Hz, 21 pulses) in EC-CA2 and SCCA1 synapses. The early-LTP induced in the presence of group III mGluR inhibitors did not transform to long-lasting LTP and was comparable to normal early-LTP in EC-CA2 and SC-CA1 synapses. Thus, the antagonist lowers the threshold for inducing LTP specifically in SC-CA2 synapses. We have now discussed these findings in the manuscript and the new data are presented in the manuscript as Figure 3â€”figure supplement 3 A-H.